# Redox Regulation of Soluble Epoxide Hydrolase—Implications for Cardiovascular Health and Disease

**DOI:** 10.3390/cells11121932

**Published:** 2022-06-15

**Authors:** Rebecca Charles, Philip Eaton

**Affiliations:** Clinical Pharmacology & Precision Medicine, William Harvey Research Institute, Queen Mary University of London, London EC1M 6BQ, UK

**Keywords:** soluble epoxide hydrolase, redox, cardiovascular

## Abstract

Cell responses to changes in their redox state are significantly mediated by reversible oxido-reductive post-translational modifications of proteins, potentially altering their activities or interactions. These modifications are important for the homeostatic responses of cells to environmental changes that alter their redox state. Such redox regulatory mechanisms not only operate to maintain health, but can become dysregulated and contribute to pathophysiology. In this review, we focus on the redox control of soluble epoxide hydrolase (sEH), which is widely expressed, including in blood vessels and cardiomyocytes. We review the different types of oxidative modifications that regulate sEH and how they may alter cardiovascular physiology and affect disease progression during stress.

## 1. Introduction

Multiple post-translational modifications (PTMs) can modify proteins and alter their stability or conformation to regulate their interactions and functional activities. Prominent examples include acetylation, phosphorylation, palmitoylation and glycosylation, together with a substantive array of oxidative PTMs (oxPTMs). Collectively, these modifications orchestrate homeostatic responses of cells; however, these mechanisms of control may be dysregulated and contribute to disease pathogenesis, including those of the cardiovascular system.

Changes in the abundance of reactive oxygen species (ROS), reactive nitrogen species (RNS) or cellular reducing biomolecules are altered during health and disease. Proteins sense these perturbations in cellular redox, with ROS or RNS reacting with amino acids to introduce oxPTMs, which can couple to a functional change that may alter physiology [1]. Significant attention has been given to the regulatory importance of the modification of cysteine, methionine and tyrosine residues in proteins, perhaps because of their greater propensity to forming oxPTMs.

Tyrosine residues can be oxidized to dityrosine and can also react with peroxynitrite to form 3-nitrotyrosine [2], which can mimic regulatory tyrosine phosphorylation [3]. Methionine residues are reversibly oxidized to methionine sulfoxide, and then further irreversibly oxidized to the sulfone [4]. The thiol groups in cysteine residues are especially sensitive to oxidation. Depending on the species and concentration of the oxidant, cysteine thiols can undergo a variety of different oxidative modifications. For example, cysteine thiols can undergo persulfidation (also known as *S*-sulfhydration), *S*-nitrosylation, *S*-sulfenylation or *S*-glutathionylation, following reactions with hydrogen sulfide, reactive derivatives of nitric oxide (so called NOx species), hydrogen peroxide or derivatives of glutathione (e.g., GSSG, GSNO), respectively [5,6,7]. Other, more stable oxidation states that cysteine residues can form include inter-protein and intra-protein disulfide bonds, as well as over oxidation reactions to sulfinic and sulfonic acids [5,6], as shown in Figure 1. Electrophilic lipids, such as 15-deoxy-Δ-prostaglandin (15d-PGJ_2_) and 10-nitro-oleic acid (10-NO_2_-OA), which we discuss in detail below, can also covalently adduct to cysteine thiols via a Michael addition reaction [8].

Multitudes of proteins are now recognised to be regulated by various oxPTMs [9,10,11,12], which can impact cellular homeostasis and disease pathogenesis. Although not discussed in this focused review, how different oxPTMs influence physiology and pathogenesis has been extensively considered elsewhere [13,14,15,16]. Mass spectrometry-based proteomics has enabled the identification and quantification of oxPTMs, which in turn has led to a greater understanding of the redox control of many different proteins and the consequences for physiology or disease progression. More recently, a large-scale study quantitatively mapped the human and mouse cysteine redox proteome in vivo [17]. This OxiMouse project identified redox networks within tissues and provided a compendium of modified proteins and mapped their often multiple sites of oxidation that in many cases demonstrated significant modification stoichiometry. It is increasingly evident that redox control of proteins is comparable with the regulation afforded by other modifications such as phosphorylation. The focus of this review is how different oxPTMs regulate the activity of soluble epoxide hydrolase (sEH), as well as discussing how these modifications affect cardiovascular physiology during health and disease.

## 2. Soluble Epoxide Hydrolase

sEH is ubiquitously expressed, including in cardiovascular-relevant tissues such as endothelial or vascular smooth muscle cells as well as cardiomyocytes, where it is an important modulator of arterial and cardiac functions [18,19,20,21,22,23,24,25]. sEH is also a susceptibility factor for human heart failure, with polymorphisms that enhance hydrolase activity increasing cardiovascular risk [26]. Conversely, inhibitors (or transgenic knock-outs) of sEH offer a broad spectrum of cardiovascular protection, including the blockade of smooth muscle proliferation [27], reduction of atherosclerosis and hypertension [18,23,28,29,30,31], prevention and regression of cardiac hypertrophy and HF [26,32,33], and fibrosis [34]. sEH is homodimeric and arranged in an anti-parallel manner, comprising an N-terminal lipid phosphatase domain and a C-terminal epoxide hydrolase domain (Figure 2A) [19]. The N-terminal domain is capable of dephosphorylating lipids such as lysophosphatidic acids (LPA) [35,36] and sphingosine-1-phosphate (S1P). The hydrolase domain catalyzes the hydrolysis of epoxy-fatty acids (EpFAs) to their corresponding diols, as shown in Figure 2 [37,38].

The active site of the hydrolase domain, as identified through biochemical and structural analysis, is located at the base of a 25 Å deep L-shaped cavity [39], where a triad of amino acids, namely Asp335, Asp496 and His524, catalyse the epoxide hydrolysis reaction [40,41]. The mechanism involves two steps, an initial ester formation between the EET and Asp335 of the enzyme followed by the water-mediated release of the diol product that is catalysed by Asp496/His524. Two tyrosines, specifically 383 and 466, provide support by establishing hydrogen bonds with the oxygen of the epoxy ring, thereby positioning it for the nucleophilic attack by the proximal aspartate [38]. These two tyrosine residues can be nitrated to inhibit the hydrolase and will be discussed in detail below.

The most widely studied substrates of sEH are epoxyeicosatrienoic acids (EETs); however, it can also hydrolyse epoxydocosapentaenoic acids (EpDPEs), α- and γ-epoxyoctadecadienoic acids (α/γ-EpODEs) and epoxyoctadecaenoic acids (EpOMEs). EETs, of which there are four regioisomers *cis* 5,6-EET, 8,9-EET, 11,12-EET and 14,15-EET, are derived from arachidonic acid. EETs have a wide range of biological activities, including dilation of systemic or coronary arteries or, conversely, constricting those of the pulmonary system. EETs are broadly pro-angiogenic, anti-inflammatory and limit platelet aggregation as well as promoting fibrinolysis. Finally, EETs also enhance cardiomyocyte contractility [42,43,44,45,46,47,48,49]. Therefore, inhibiting sEH pharmacologically elevates its substrates such as EETs, allowing them to exert their beneficial actions. Indeed, inhibiting sEH has identified the hydrolase as an important modulator in cardiovascular and renal health, with additional roles in nociception [19,50,51].

## 3. Oxidative Post-Translational Modifications of sEH

Until 2009, little was known about how sEH activity was regulated, and it was thought to be principally determined by its expression abundance [52]. However, it is now apparent that a number of different oxPTMs, which are discussed below, as are their effects on cardiovascular health and disease (summarized in Table 1), regulate this hydrolase.

### 3.1. Tyrosine Nitration

Barbosa-Sicard et al. showed that sEH is inhibited by tyrosine nitration (Figure 2B), providing perhaps the first evidence that sEH could be oxidized by a PTM. Y383 and Y466, key amino acids for enzymatic sEH activity, were nitrated by peroxynitrite (ONOO^−^) or the ONOO^−^ generator 3-morpholinosydnonimine (SIN-1) [53]. Tyrosine nitration at either of these sites resulted in hydrolase inhibition. Streptozotocin, a compound used to induce diabetes but interestingly also contains an N-nitroso moiety that may donate nitric oxide, was found to increase tyrosine nitration. Traditionally, tyrosine nitration has been associated with damage and disease. This may be due to the widely held ‘belief’ that oxidants are principally harmful, and indeed this may be the case with peroxynitrite because its high reactivity provides potential for non-selective oxidations. However, in this scenario, inhibiting sEH by tyrosine nitration would be expected to increase cardioprotective EETs, which may mitigate against dysfunction arising from other damaging modifications.

### 3.2. Adduction of Electrophilic Lipids to sEH

We found 15d-PGJ_2_ adducts to and inhibits the activity of sEH (Figure 2C) [54]. 15d-PGJ_2_ is an endogenously generated electrophile that contains two α,β-unsaturated carbonyls that enable it to react with nucleophilic thiol moieties in proteins via Michael addition reactions (Figure 2C). As C521 (mouse protein sequence numbering) is conserved and proximal to the catalytic triad, as discussed above, it was identified as a likely candidate to mediate the electrophilic lipid and subsequent inhibition. Although other targets of 15d-PGJ_2_ have been identified, with perhaps the best characterised being the nuclear peroxisome proliferator-activated receptor γ [55], we demonstrated a causal link between this electrophilic lipid and its ability to adductively inhibit sEH to induce coronary vessel vasodilation [54]. This electrophilic mechanism of inhibition also contributed to coronary artery hypoxic vasodilation [54]. Subsequently, other electrophilic lipids, such as 10-NO_2_-OA, were also found to adduct to sEH C521 to inhibit hydrolase activity (Figure 2D) [56,57]. 10-NO_2_-OA is known to protect against myocardial infarction, preserve left ventricular function [58], dilate arteries [59], attenuate platelet activation and reduce inflammation via cyclic guanosine monophosphate (cGMP)-independent mechanisms [60,61]. Perhaps these protective actions, such as the reduction in blood pressure and reducing myocardial infarct size, may also be explained by sEH inhibition. Given the complexities of establishing if the blood pressure lowering effects of 10-NO_2_-OA were due to sEH inhibition, a transgenic knock-in mouse in which C521 was systemically replaced with a serine was generated. This enabled investigations to determine whether electrophilic lipids exert their blood pressure lowering actions causally by inhibiting sEH. These knock-in mice were studied in the context of angiotensin II-induced hypertension, measuring their blood pressure lowering response to 10-NO_2_-OA compared to wild-type controls. The nitro-fatty acid inhibited sEH in wild-type mice in which EETs concomitantly accumulated and so explains the blood pressure-lowering observed. This lowering of blood pressure was completely absent in the knock-in mice that lack the cysteine to which the nitro-lipid adducts to cause hydrolase inhibition. Interestingly, inhibition of sEH by 10-NO_2_-OA highlighted a mechanism to account for the protection from hypertension afforded by the Mediterranean diet. Paradoxically, people who consume this diet, which is rich in fats and therefore is anticipated to promote cardiovascular disease, are in fact protected from such conditions [62,63]. The formation of nitrated fatty acids such as 10-NO_2_-OA are promoted under acidic conditions in the stomach following consumption of green leafy vegetables with nitrite and unsaturated fats that are key components in the Mediterranean diet [64,65,66]. Subsequent studies identified C423 as an additional site for electrophilic lipid adduction [57]. Although this cysteine is not present in the murine orthologue, adduction of a lipid electrophile to this site in recombinant human sEH was inhibitory. However, this inhibitory mechanism has yet to be substantiated in cells or in vivo. Electrophile adduction at C423 is notable because it is distal to the active site and can therefore be described as allosteric. Therefore, targeting C423 may be useful in the development of allosteric sEH inhibitors [57]. C521 is located at the exit of the hydrophobic tunnel and therefore may not be completely allosteric due to its proximity to the active site. However, the mechanism of inhibition at C521 could potentially be exploited by the development of novel electrophilic drugs to combat cardiovascular disease. Collectively, our work clearly demonstrates that the enzymes’ activity can be inhibited by electrophilic lipids.

### 3.3. S-Nitrosation of sEH

In 2016, Ding et al. showed sEH is activated by *S*-nitrosation (Figure 2E) [67]. During myocardial reperfusion, sEH was activated and lead to a loss of cardioprotective EETs. A *trans*-nitrosating nitric oxide donor, namely nitrosocysteine (CysNO), recapitulated this activation of sEH in vitro (Figure 2E). C141 was identified as the main site of *S*-nitrosation, although a number of different cysteine residues were also found to be modified in this way, including C264, which will be considered further below. Interestingly, C141 is located in the N-terminal domain of sEH, however due to the domain swapped anti-parallel architecture of sEH, a SNO modification at this site perhaps stabilises bonds in the hydrolase domain of the opposite monomer, leading to enhanced hydrolase activity. *S*-Nitrosoglutathione (GSNO) is another nitric oxide donor that might be expected to activate sEH in a similar manner as described here by CysNO. These authors also reported that GSNO similarly activated sEH, whereas we found that it did not do this [54]. It is possible that GSNO may induces S-glutathionylation of sEH and the effect of this disulfide modification on activity remains unclear.

### 3.4. Intra-Disulfide Formation in sEH

In 2021, we described a mechanism by which sEH can be activated by oxidative intra-disulfide formation (Figure 2F) [68]. Oxidation induced either by exogenously applied hydrogen peroxide or the pro-oxidant vasopressor hormone angiotensin II increased sEH activity by catalyzing a disulfide between Cys262 and C264 (mouse protein sequence numbering), as depicted in Figure 2D. This oxidative activation increased the hydrolysis of vasodilatory EETs. Indeed, following angiotensin II treatment, the EET/DHET ratio decreased, consistent with and potentially explaining, at least in part, the pressor action of angiotensin II. This is notable, as we have previously shown that *S*-nitrosated proteins generically transition to form a disulfide bond [69]; hence, it is likely that the *S*-nitrosation of C264 observed by Ding et al. induced the activating intra-disulfide with C262.

## 4. Redox Regulation of sEH in Blood Vessels

Decreasing sEH activity, either pharmacologically or genetically, has illustrated a broad role for this hydrolase and its EET substrates in vascular homeostasis. EETs are angiogenic, anti-inflammatory, and suppress platelet aggregation [45,47,70,71]. Regulation of angiogenesis is critical to the maintenance of vascular and myocardial health, with EETs able to increase endothelial cell proliferation to promote such vascularization [45,72,73]. On this basis, inhibiting sEH, because of the angiogenic roles of EETs, promotes neovascularization in ischemic tissue. Interestingly, the redox status of the cell affects angiogenesis [74,75]. Consequently, oxidative modifications that inhibit sEH, may be expected to promote angiogenesis. Although not directly investigated in the study reporting that tyrosine nitration of sEH is inhibitory [53], it is conceivable that this oxPTM enables nitrosative signaling to be coupled with angiogenesis. Our group demonstrated, albeit in a cancer model, that sEH inhibition induced by nitro-oleate adduction to C521 promoted angiogenesis [76]. Indeed, this electrophilic inhibition of sEH increased cell proliferation and migration in wild-type (WT) mice more than in C521S knock-in mice lacking the critical cysteine that renders the hydrolase electrophile-resistant. In direct contrast to sEH inhibition promoting angiogenesis, increasing hydrolase activity is anticipated to reduce it. Therefore, following *S*-nitrosation or disulfide bond formation, namely the scenarios mentioned above that activate sEH, a decrease in cell proliferation and migration may be expected. However, treatment with a diol restored blood flow in sEH null mice to levels observed in WT mice during hind limb ischemia [77]. Similarly, sEH null mice displayed attenuated sprouting angiogenesis in their retinas, which could again be rescued by the application of the diols [78]. Therefore, in these scenarios, increasing sEH activity by either *S*-nitrosation or disulfide formation may be beneficial. While it is clear that sEH plays a role in angiogenesis, further research is required to elucidate how sEH is regulated to elicit its opposing angiogenic effects in different tissues and the precise roles redox regulation play in specific cells or scenarios.

The anti-inflammatory actions of EETs also contribute to vascular homeostasis. 11–12 EET decreased the number of adherent and rolling mononuclear cells in mice following TNF-injections [79]. Similarly, sEH null mice or in AUDA-BE treated mice had an attenuated inflammatory response to lipopolysaccharide compared to WTs [80]. Likewise, sEH inhibition not only decreased infiltration of macrophages in angiotensin II-induced or deoxycorticosterone (DOCA) salt hypertension, but also attenuated renal inflammation [81,82]. Despite evidence that sEH inhibition is anti-inflammatory, more research is needed to establish whether the sEH redox state is modulated in such scenarios, which is likely given the roles of oxidants in inflammation [83]. As tyrosine nitration and electrophilic lipid adduction both inhibit sEH, it is conceivable that each of these mechanisms increase EETs and their anti-inflammatory actions—a prospect that warrants further investigations.

EETs are vasodilators that induce vascular smooth muscle cell (VSMC) hyperpolarization and the activation of large-conductance calcium activated potassium channels (K_Ca_) to cause arterial dilation [47,84]. Some evidence indicates that EETs can also activate Ca^2+^-induced intermediate-conductance (IK_Ca_) and small-conductance (SK_Ca_) potassium channels that may also participate in cell hyperpolarization [85], which spreads via gap junctions [47]. EETs can also activate transient receptor potential vanilloid channel 4 on VSMCs [86]. EETs released by endothelial cells are also thought to activate a yet unidentified EET receptor on VSMCs, leading to adenylate cyclase and protein kinase A activation, activating BK_Ca_ and ATP-sensitive potassium (K_ATP_) channels to induce hyperpolarization and vasodilation [19]. Based on the vasodilatory actions of ETTS, inhibiting the hydrolase either through pharmacological or genetic manipulation decreased blood pressure in several different animal models of hypertension [30,31,81,87,88]. As discussed above, we found that the electrophilic lipid NO_2_-OA, covalently adducted to sEH at C521 to inhibit it, increased EETs/DHETs, thereby lowering blood pressure in angiotensin II-induced hypertensive mice. This study clearly demonstrates a role for the redox control of sEH in blood pressure regulation and the pathogenesis of hypertension. It is plausible that other mechanisms of inhibitory sEH oxidation, such as tyrosine nitration, may also couple to vasodilation. On the other hand, oxidative modifications that activate the enzyme are anticipated to have the opposite effect and induce vasoconstriction instead due to the increased hydrolysis of vasodilatory EETs. Interestingly, the vasopressor angiotensin II not only increases cellular H_2_O_2_ [89,90] but also decreases EETs in mice, consistent with sEH being oxidized in this scenario [68].

## 5. The Role of Redox Regulation of sEH in the Heart

Myocardial sEH has been significantly investigated in terms of cardiac pathophysiology, with relatively few studies investigating its role in physiological functions during health. However, similar to the dilation of the systemic vasculature, EETs can also relax the coronary vasculature by activating K_Ca_ channels [91]. Our work with 15d-PGJ_2_ also demonstrated that EETs relax the coronary vasculature, whereby the electrophilic lipid induced a prominent and sustained vasodilation in ex vivo perfused rat hearts. This coronary vasodilation was due to 15d-PGJ_2_ adducting to C521 and inhibiting sEH, a mechanism that contributed to coronary hypoxic vasodilation [54].

In myocytes, 5,6-EET or 11,12-EET increased cell shortening and intracellular calcium concentrations [92]. Other studies have since demonstrated that EETs can inhibit cardiac myocytes Na^+^ channels [93], activate L-type Ca^2+^ channels [94], as well as increase K_ATP_ channel openings [95]. Despite these effects on cardiac myocyte ion channel activity, very few investigations have explored EETs action on normal cardiac function, perhaps because of the difficulty of functionally separating the contractile actions of EETs from their coronary vasodilatory actions [47]. Further research is required to establish if there is a role for redox control of sEH in these processes. However, because oxPTMs that inhibit sEH will likely increase EETs, they may reasonably be anticipated to increase cardiac myocyte contraction.

The role of myocardial sEH in cardioprotection has been widely studied. Hearts were protected from cardiac hypertrophy as well as myocardial ischemia and reperfusion injury when the hydrolase was absent in sEH null mice or was pharmacologically inhibited in WTs [33,88,96]. Cardioprotection has been attributed to the EETs that are increased when sEH is absent in knockout animals or inhibited pharmacologically. In terms of redox regulation of sEH and cardioprotection, electrophilic lipid adduction to C521 in sEH alleviated cardiac hypertrophy induced by angiotensin II [56]. Although this protection from hypertrophic growth is likely explained by the blood pressure-lowering action of the lipid, a contribution from the inhibition of cardiomyocyte sEH is also plausible. Our observation is consistent with other studies showing that classical sEH inhibitors not only prevented maladaptive growth during cardiac pressure load in rodents, but in some cases reversed established hypertrophy [33,96].

Administering 10-NO_2_-OA to hearts prior to ischemia and reperfusion protected them from loss of contractile function [97], which is likely to be partially mediated by it inhibiting sEH and so increasing EETs that have established cytoprotective actions. Although sEH inhibition via either electrophilic lipid adduction or tyrosine nitration may be anticipated to protect against myocardial ischemia and reperfusion injury, further research is required to determine this.

Interestingly, following angiotensin II-induced cardiac hypertrophy or ischemia and reperfusion injury, sEH activity is increased [33,98]. This may be expected as sEH expression is increased upon angiotensin II treatment [98], as well as after ischemic injury [99]. However, this increased expression did not fully explain the increase in hydrolase activity, consistent with the prospect of oxidative activation [67,68]. During hypertrophy or ischemia and reperfusion, ROS are increased as NADPH oxidases and xanthine oxidase are stimulated, producing superoxide and H_2_O_2_ [90,100,101]. These oxidants may account for the increased hydrolase activity observed, considering that we have demonstrated that peroxide molecules activate sEH [68].

## 6. Perspectives and Future Directions

It notable that the Oximouse compendium shows that a number of cysteine residues in sEH, both in the human and mouse proteome, are susceptible to oxidation, and these are altered during aging in various different tissues [17]. The Oximouse dataset demonstrated that there is not an overall increase in protein oxidation with age, but instead that the redox networks are remodeled [17]. In fact when looking at sEH in this dataset, it is apparent that some cysteines become more oxidized during aging, whereas others are concomitantly reduced. Although the number of these cysteines have previously been identified as sites of oxidation, including C521, C141 and C423, further research is required to elucidate the functional significance of the other sites identified.

For a protein to be considered truly redox regulated, the oxPTMs need to be able to be reversed as this enables dynamic, two-way control. In terms of the oxPTM in sEH discussed above, one wonders about their reversibility and if they are enzymatically reversed. In the case the intra-protein disulfide in sEH, we provided evidence that thioredoxin reduces the disulfide back to the basal reduced state [68]. S-nitrosylated proteins can supposedly be reversed by thioredoxin [102], however this has yet to be determined specifically for sEH. Although not conclusive, as its identity remains elusive, there is evidence for a cellular denitrase that reverses protein tyrosine nitration [103]. Such a denitrase may reverse the inhibitory modification at Y383 and Y466. Formation of the covalent protein modification with either 15d-PGJ2 or NO2-OA and sEH was thought to be irreversible, with proteolysis being the main mechanism of reversal. However, there is now evidence that electrophilic nitroalkylation is reversible and may be mediated by GSH [104].

Interestingly, the studies describing covalent adduction of endogenous electrophilic lipids offers some hope for the generation of a novel class of sEH inhibitors which have advantages over classical inhibitors [105]. The sites of this endogenous inhibitory mechanism may be exploited for the design of novel drugs, which may be used to target hypertension and other cardiovascular dysfunction. Covalent compounds have traditionally been avoided in the development of therapies, as they were thought to non-selectively conjugate with many different proteins. However, this view has been challenged by studies showing that covalent compounds can offer selectivity and potency benefits over traditional drugs [106,107,108,109]. Indeed, there are a number of widely used drugs, such as penicillin and aspirin [107], which mediate their therapeutic actions via covalent adduction to target proteins.

Previously, antioxidants have been tested as a therapy for cardiovascular diseases in an attempt to combat excessive oxidant production. However, despite multiple clinical trials assessing their efficacy, the outcomes have been disappointing, with no cardiovascular protection observed, and in some instances they were even harmful [110,111]. However, these studies failed to consider the homeostatic and adaptive oxidant signaling that may be lost with antioxidant interventions. Therefore, by understanding specific actions of different ROS or RNS, it may be possible to target precise oxidation events with antioxidants. Interestingly, certain antioxidants can paradoxically oxidize proteins. For example, the antioxidant resveratrol can convert to a reactive quinone and then oxidize Protein Kinase G Iα [112]. How antioxidants influence sEH activity is worthy of exploration.

This review has primarily focused on the hydrolase domain of sEH. However, as mentioned above, sEH also contains an N-terminal lipid phosphatase domain [113,114] which can dephosphorylate LPA [35,36] and S1P. As LPA and S1P are important modulators of vascular homeostasis, the redox regulation of the N-terminal phosphatase may also play a role in the context of cardiovascular health and disease. Interestingly, Oximouse compendium indicates that the cysteine residues within the N-terminal domain can be redox modulated [17]. Likewise, there are different isoforms of epoxide hydrolase, including microsomal epoxide hydrolase, epoxide hydrolase 3 and epoxide hydrolase 4. Although minimally expressed in cardiac cells, it would be interesting to determine if they are redox controlled.

## 7. Closing Thoughts and Considerations

It is now abundantly evident that many proteins are regulated by oxPTMs that couple changes in cellular redox status to functional responses of proteins and the cells they are found in. Over the last 20 years or so, we have increasingly changed our view of the role of oxidants in cells and tissues. Although they have important roles in dysregulation and disease pathogenesis, there is a plethora of evidence for their regulatory roles during health and adaptive changes that limit disease progression. It is evident that the redox regulation of cardiac and vascular sEH is highly intricate, and much remains to be understood. Importantly, redox control of the hydrolase contributes both to protective as well as maladaptive pathways, and this likely depends upon the specific ROS source and their concentration and locality to sEH, as shown in Figure 3.

## Figures and Tables

**Figure 1 cells-11-01932-f001:**
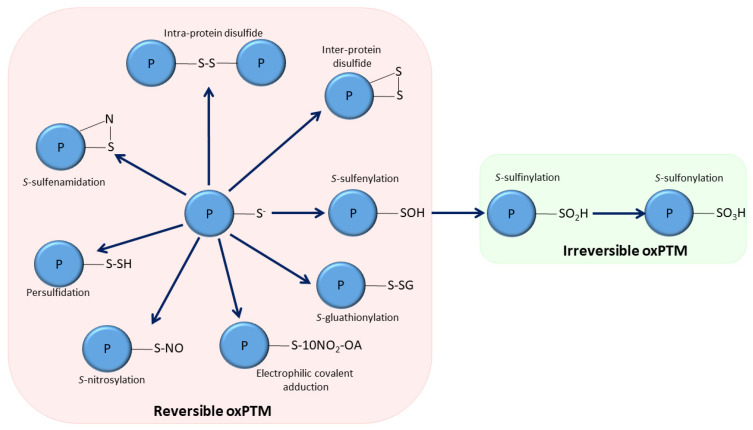
Summary of the oxPTMs formed in protein thiols. Thiolate anions can react with a variety of reactive oxygen or nitrogen species to form reversible (intra-protein disulfides, inter-protein disulfides, *S*-sulfenylation, *S*-nitrosylation, *S*-glutathionylation, persulfidation, *S*-sulfenamidation, electrophilic covalent adduction) and irreversible hyper-oxidized (*S*-sulfinylation, *S*-sulfonylation) redox states.

**Figure 2 cells-11-01932-f002:**
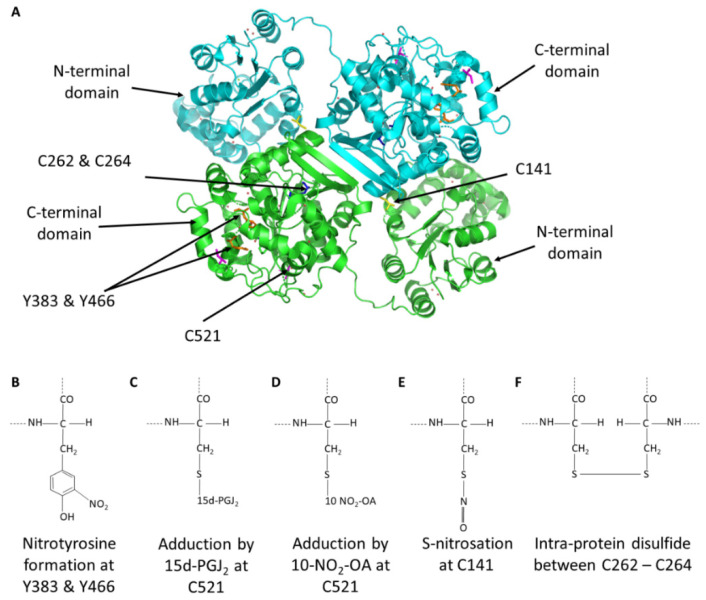
sEH structure (**A**) Domain swapped homo-dimeric structure of sEH. (**B**–**F**) Overview of various oxidative modifications of sEH and the sites at which they occur.

**Figure 3 cells-11-01932-f003:**
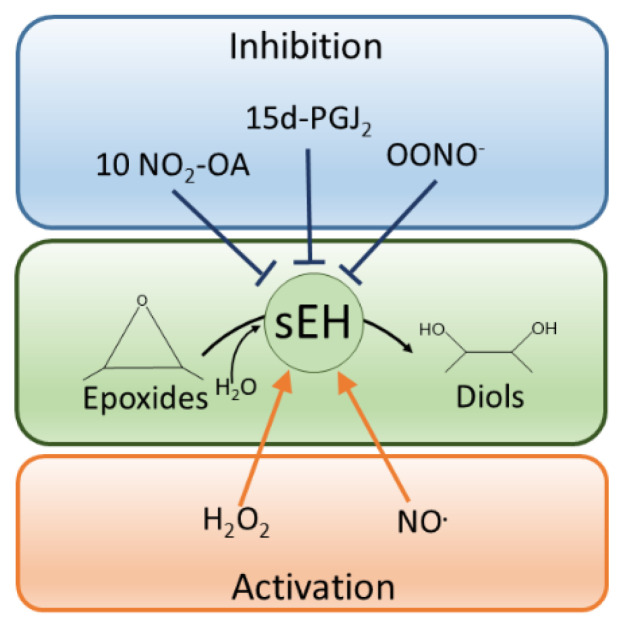
sEH activity is regulated by diverse redox mechanisms. Different reactive oxygen species or reactive nitrogen species can differentially modify sEH to alter its activity. For example, the lipid electrophiles 10-NO_2_-OA and 15d-PGJ_2_ as well as peroxynitrite inhibit the hydrolase, whereas hydrogen peroxide or NO stimulate it.

**Table 1 cells-11-01932-t001:** Summary of different oxPTM that modulate sEH activity and their influences on cardiovascular health and disease.

oxPTM	sEH Activity	Potential and Known Effects on Cardiovascular Function
Tryosine nitration	Inhibited	Angiogenic Anti-inflammatory Cardioprotective Anti-hypertensive Coronary vasodilatory Increase myocyte contraction
Electrophilic lipid adduction	Inhibited	Angiogenic Anti-inflammatory Cardioprotective Anti-hypertensive Coronary vasodilatory Increase myocyte contraction
*S*-nitrosation	Activated	Anti-angiogenic Vasoconstriction Hypertrophic Increased ischemic & reperfusion injury
Intra-disulfide Formation	Activated	Anti-angiogenic Vasoconstriction Hypertrophic Increased ischemic & reperfusion injury

## Data Availability

Not applicable.

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
