# Peer review of "Redox Regulation of Soluble Epoxide Hydrolase—Implications for Cardiovascular Health and Disease"

_cells, 2022, doi:10.3390/cells11121932_

Round 1

Reviewer 1 Report

This is a well-written review article by Charles R. et al. outlining recent studies on the redox control of soluble epoxide hydrolase and its involvement in cardiovascular health and disease, which the reviewer believes would be of interest to a broad audience of Cells readers. This is a well-organized review article overall. There are seven sections in this review article:

In Section 1, the authors introduced the oxidative posttranslational modifications of tyrosine, cysteine, and methionine residues within target proteins, with a focus on cysteine oxidation; mass spectrometry-based proteomics is a major approach to studying OPTM. In line 31, Is “nitro-oxidant species” referred to reactive oxygen and nitrogen species? Please clarify it. It is suggested to expand the introduction about the ways of OPTM modulating target proteins and how they are involved in cellular homeostasis and pathogenesis of disease in general. The authors may need to discuss other regulatory mechanisms of sEH in health and diseases.  

In Section 2 and 3, the authors introduced soluble epoxide hydrolase, and summarized major OPTMs that occur on their cysteine residues. In 3.1 subsection, the authors may elaborate the discussion about the tyrosine nitration of sEH by peroxinitrite, which is known as a major pathogenic oxidant species. However, sEH activity is inhibited by this OPTM and expected to be beneficial to cardiovascular disease.

In Section 4 and 5, the authors nicely discussed the redox regulation of sEH in blood vessels and heart, focusing on  the cardiovascular functions of its substrate EETs including proangiogenic, anti-inflammatory, vasodilatory effects. It is suggested to add a table or diagram to summarize these research findings, thus providing the reader with the information at one glance. Also, the authors need to summarize the studies and discuss the possible relationship between sEH activity and cardiovascular oxidative stress.

In Section 6 and 7, the authors provided new perspectives, such as the possibility of using covalent adduction of endogenous electrophiles to generate a novel class of sEH inhibitors for cardiovascular diseases, including hypertension. The authors should discuss the target specificity of this therapeutic strategy, and speculate the potential role of antioxidants in regulating sEH activity. 

Reviewer 2 Report

This is a manuscript written by authors that have made significant contributions to identifying translational mechanisms regulating the activity of the soluble epoxide hydrolase. In this review article the authors summarise their previous work and place it in the general context of epoxide and diol biology. Perhaps one oxidative PTM the authors did not address was sulfhydration which has been the focus of several recent studies.

The text is rather heterogeneous with some sections being really well written and others being rather “bumpy”. This can be simply resolved by a bit of editorial care.

Abstract: not capitalise epoxide hydrolase in line 13.

A better abbreviation for oxidative post-translational modification would be oxPTM - as used in other reviews on the topic.

On page 1 (lines 27 to 37) the first paragraph could be rewritten to avoid duplication regarding the consequences of oxPTM on protein function.

Lines 321 onwards: the last page of the review seems a little bit late to mention the lipid hydrolase domain.

A scheme should accompany the description of the oxidative PTMs described in paragraph 3 (page 1).

Page 2, line 49 the D in Deoxy should be small.

The wording of the last sentence in the introduction (page 2, line 61 to 63) could be better formulated.

Section 2: the first sentence (lines 65 to 67) contains one too many “including”.

The figures are positioned too far away from their first callout in the text.

Line 172 - 173: “can is” requires correction.

Line 200 to 202: The lack of recovery of blood flow following hindlimb ischaemia in the SEH null mice shouldn’t really be included in the discussion of angiogenesis (see reference 70) as the defect here was related to a defect in stem and progenitor cell proliferation as well as bone cell mobilisation.
